# The landscape of alternative splicing in granulosa cells and a potential novel role of YAP1 in PCOS

Linlin Yang[1‡], Jianhua Chen[2‡], Hui Miao[1], Na Li[1], Huilin Bi[1], Ruizhi Feng[2,3,4]*, Congxiu Miao[1]*

1 Department of Reproductive Genetics, Key Laboratory of Reproduction Engineer of Shanxi Health Committee, Heping Hospital of Changzhi Medical College, Changzhi, China, 2 State Key Laboratory of Reproductive Medicine and Offspring Health, Nanjing Medical University, Nanjing, China, 3 Reproductive Medical Center of Second Affiliated Hospital of Nanjing Medical University, Nanjing, China, 4 Innovation Center of Suzhou Nanjing Medical University, Suzhou, China

‡ LY and JC are contributed equally to this work and share the first authorship.
* ruizhifeng@njmu.edu.cn (RF); mcxms@163.com (CM)

## Abstract

Polycystic ovary syndrome (PCOS) is a prevalent yet complex reproductive endocrine disorder affecting 11–13% of women worldwide. Its main symptoms include elevated androgen levels, irregular menstrual cycles, and long-term metabolic and offspring health implications. Despite the disease's multifaceted nature involving genetic, epigenetic, and environmental factors, the role of alternative splicing in ovarian granulosa cells remains relatively unexplored. This study aims to investigate the transcriptional and alternative splicing characteristics of granulosa cells in PCOS patients and to elucidate the potential functional consequences of these changes. Analysis of previous published transcriptome sequencing data identified 491 upregulated and 401 downregulated genes in granulosa cells of PCOS patients, significantly involved in immune-related processes. Additionally, 1250 differential splicing events, predominantly involving exon skipping and affecting 947 genes, were detected. These genes with alternative splicing patterns were found to be enriched in endoplasmic reticulum stress and protein post-translational modification processes, suggesting their role in PCOS pathology. Moreover, the study highlighted that the utilization of different splice isoforms of the YAP1 gene may impact its interaction in the Hippo signaling pathway, influencing the pathogenesis of PCOS. These findings underscore substantial alterations in alternative splicing in granulosa cells of PCOS patients, providing a novel viewpoint for comprehending the molecular underpinnings of PCOS and suggesting potential avenues for therapeutic intervention.

## Introduction

Polycystic ovary syndrome (PCOS) is one of the most common reproductive endocrinological disease, influencing about 11–13% women worldwide [1], whose etiology has not been fully

**Data Availability Statement:** The raw RNA-seq data published in previous article can be accessed using the following accession number GSE138518 in GEO database (https://www.ncbi.nlm.nih.gov/geo/query/acc.cgi). All relevant scripts and

resources associated with this article have been made publicly available on GitHub. They can be accessed at the following link: https://github.com/Jenny-chen98/ASinPCOS/tree/main.

**Funding:** This research was financially supported by grants from National Natural Science Foundation of China (81971451, 31900605), Innovative and Entrepreneurial Team of Jiangsu Province (JSSCTD202144), Innovative and Entrepreneurial Talent Program of Jiangsu Province to Ruizhi Feng, and Natural Science Foundation of Shanxi Province (NSF201901D111325) to Congxiu Miao. The funders were involved in the study design, manuscript preparation and manuscript review.

**Competing interests:** The authors have declared that no competing interests exist.

understood yet. According to the latest 2023 ASRM guidelines, the main symptoms of PCOS are hyperandrogenism and irregular cycles [2]. Previous studies mostly focused on the endocrine therapy together with treatment of assisted reproductive technology to solve the problem of potential subfertility of PCOS, more and more recent researchers paid attention to the long-term impact of this disease such as metabolic syndrome, higher risk of diabetes mellitus and cardiovascular disease [3–5]. Transgenerational effects such as metabolic dysfunction [6], psychiatric and mild neurodevelopmental disorders [7] even in male offsprings have been reported, indicating a systemic transmitting implication [8]. Multiple factors contribute to the pathogenesis of PCOS, including genetic predisposition, epigenetic changes, gut microbiota, environmental impacts and life style [9], however there are still large fraction of its regulatory mechanisms to be fully uncovered.

The core symptom of PCOS is the dysfunction of hormone homeostasis, and the most significant part is the transition of androgen producing from thecal cells which wrap the whole follicle to estrogen, taking place in granulosa cells (GCs), the somatic cell around the oocyte [10]. The abnormality of ovarian granulosa cells is the major origin of the morphological change in PCOS showing polycystic ovary in ultrasonography scanning. It has been reported that in PCOS patients, the thecal cells showed hyperplasia and the granulosa cells exhibit thin layers [11], leading to infrequent ovulation. Evidence also showed abnormal cell proliferation, apoptosis and metabolism happened in GCs of PCOS patients [12], with involvement in mitochondrial function such as oxidative phosphorylation, ATP production and ROS clearance [13, 14]. The molecular mechanisms of PCOS pathology are complicated, involving the regulation of various levels, including DNA, epigenetic modifications, transcriptional and translational factors, metabolites and environmental issues [3, 15–18].

Transcriptional regulation is the first step of genetic and epigenetic factors influencing the downstream biological process in biological organisms. Spatiotemporal gene expression is the fundamental element of life; however, it is not limited in the amount or location of a specific gene expressing, complicated forms of transcriptional regulation enrich the accuracy and fine tune the molecular process on the RNA level. Alternative splicing (AS) is one important but less well-studied approach of transcriptional regulation which contributing to numerous human disease such as cancer, neurological / autoimmune / endocrine system / digestive / cardiovascular diseases [19]. A large number of *cis*-elements such as enhancers and *trans*-elements such as relative RNA binding proteins are involved in this complicated process, resulting in different kinds of AS events. Among them, the alternative 5'/3' splice site (A5SS or A3SS), retention intron (RI), mutually exclusive exon (MXE) and exon skipping (ES) are most commonly seen events that produce different RNA isoforms and usually different proteins. Certain AS events with a specific protein product may directly trigger or decide an essential biological trajectory [20], and global AS status can serve as a signature for diagnosis / prognosis [21–23] or reflecting potential intrinsic characteristics of many diseases even future effective pharmaceutical target [19].

In the current study, we aim to generate a comprehensive landscape of alternative splicing in PCOS and investigate the potential key AS events that could possibly contribute to the etiology and pathology of this complicated disease. We reanalyzed the global expression of a previously published dataset with the RNA sequencing of granulosa cells in PCOS and control samples [24]. We found that there are significant transcriptional and alternative splicing differences in granulosa cells of PCOS patients, involving immune-related processes and endoplasmic reticulum stress. The main type of splicing event is skipping exons, especially the splicing changes of YAP1 gene may affects its role in the Hippo signaling pathway. Our findings provide a deeper understanding of the complexity of PCOS and may bring new insight into future basic research and potential therapy development.

## Results

### Transcriptional profiles of human PCOS and normal granulosa cells

We obtained transcriptome sequencing data of granulosa cells from both PCOS patients and normal donors from previous studies [24]. In brief, mature follicles free of any visible blood contamination were harvested from participants (PCOS and controls) aged 20–35 years who had undergone controlled ovarian hyperstimulation. Granulosa cells were then isolated by centrifugation for transcriptomic analysis. After removing adapters and filtering out low-quality reads, we mapped the clean reads to the human genome and found that an average of 95% of reads were specifically mapped (S1A Fig). The results of principal component analysis (PCA) of FPKM matrix of all expressed genes revealed a clear distinction between the two groups (S1B Fig).

Next, we examined the differentially expressed genes (DEGs, |log2FoldChange| $> 2$ and adjusted p-value $< 0.05$) between the two groups. We plotted these genes on a volcano plot (Fig 1A) and identified 491 up-regulated genes and 401 down-regulated genes, out of which 352 and 171 genes were protein-encoded, respectively (see Fig 1B). When we conducted Gene Ontology (GO) enrichment analysis on all the expressed genes (defined as those with at least one mapped read in any sample), we identified a total of 2618 entries in the biological process category. The most frequent cluster identified was related to RNA metabolism and splicing (Fig 1C). Additionally, the differentially expressed genes (DEGs) showed a significant enrichment in immune-related terms, where the frequency of immune-related terms is notably higher than other categories (S2 Fig).

### Dramatic changes in alternative splicing (AS) of human PCOS and granulosa cells

Alternative splicing contributes to protein diversity, and aberrant splicing events are often indicative of pathogenic outcomes. Given the strong correlation between RNA splicing and the terms for all gene enrichment described earlier, we first analyzed differential alternative splicing events (DASEs, |ΔIncLevel|$>0.1$ and FDR $< 0.05$) between the PCOS and normal groups using rMATS (replicate Multivariate Analysis of Transcript Splicing) program [25]. Five types of alternative splicing events can be identified by rMTAS, namely skipped exon (SE), alternative 5' splice site (A5SS), alternative 3' splice site (A3SS), mutually exclusive exons (MXE) and retained intron (RI). A total of 1250 differential splicing events were detected in 947 genes, suggesting that at least one differential splice event was present in each gene (see S1 Table). The upset plot revealed that a single gene could contain up to four different types of DASEs. Statistical analysis of the different splicing event types showed that SE events accounted for 49.2% of all splicing events, followed by MXE events, which comprised 19.68% (Fig 2A). Additionally, we identified 17 genes exhibiting more than three alternative splicing events, four of which were long non-coding RNAs (Fig 2B). SNHG5 had the highest number of splicing events, and it has been found to regulate follicle growth in PCOS, making it a potential therapeutic target [26]. The observed differential splicing events in PCOS suggest its potential involvement in PCOS development.

### Genes with altered AS patterns are specifically associated with endoplasmic reticulum stress

To systematically summarize the impact of AS events alteration on gene products and function, we conducted K-means clustering of differentially spliced genes in rMATS based on their expression levels. Subsequently, we carried out GO enrichment analysis for each gene class.

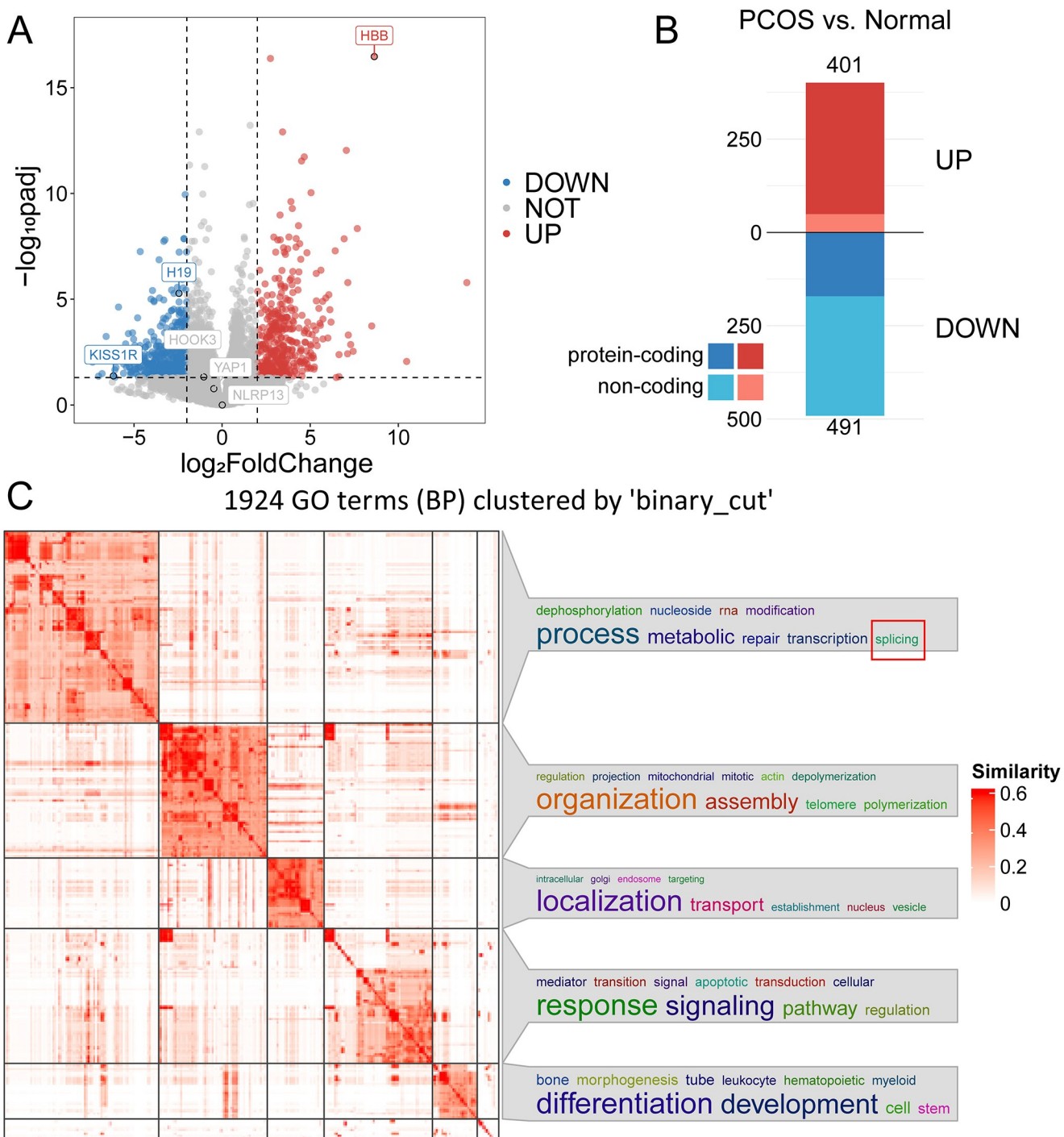

**Fig 1. Characteristics and differences between human Normal and PCOS granulosa cell transcriptomes.** (A) A volcano plot depicting significantly up-regulated or down-regulated genes between the two groups. (B) Compared with the Normal group, 401 and 491 genes were up-regulated (shown in red) and down-regulated (shown in blue) in PCOS granulosa cells, respectively. Dark colors represent protein-coding genes and light colors represent non-coding genes. (C) The cluster of BP terms is enriched based on all expressed genes (defined as those with at least one mapped read in any sample). Enrichment terms were clustered based on semantic similarity, and a summary of the biological functions within each cluster was visualized as word clouds, which are attached to the right side of the similarity heatmap.

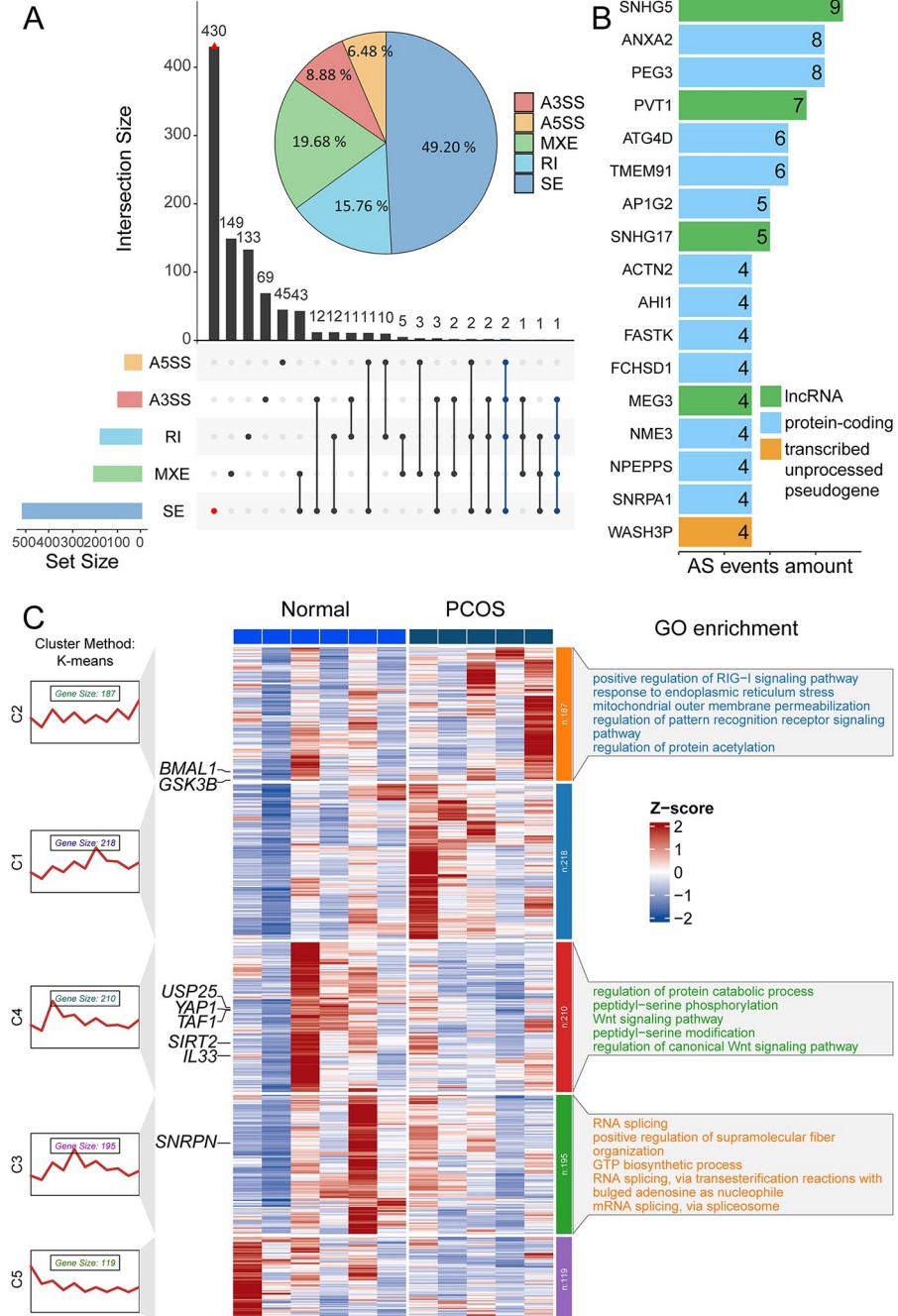

**Fig 2. Changed differential alternative splicing events (DASEs) patterns between PCOS and Normal granulosa cells.** (A) Statistics of detailed DASEs from rMATS. The upset plot illustrates the types of differential splicing events present in each gene. The height of the upper black bars in the upset plot corresponds to the number of gene sets with highlighted black dots in each column. The colored bars on the right indicate the number of genes associated with each differential splicing event type. The pie chart, on the other hand, shows the percentage of differential alternative splicing events. A3SS, alternative 3' splicing site; A5SS, alternative 5' splicing site; MXE, mutually exclusive exons; RI, retained intron; SE, skipped exon. (B) Genes with greater than 3 DASEs. Different colors represent different biological types of genes. (C) Cluster and GO enrichment of genes that contained differential alternative splicing. The K-means algorithm was used for clustering. To ensure the reproducibility of the results, we set the random seed to 2024 and determined the optimal number of clusters to be 5 based on the elbow method. As a result, the FPKM expression matrix of genes with differential splicing was subsequently clustered into 5 groups for further analysis.

We identified a total of 5 categories, with 3 of them showing significant enrichment. Notably, class C2 comprised 187 genes enriched for endoplasmic reticulum stress (ERS) and protein post-translational modifications, while class C4 contained 210 genes enriched for pathways related to Wnt signaling and protein metabolism. The enrichments in C3 were largely related to RNA splicing regulation (Fig 2C). Multiple studies have demonstrated the involvement of ERS in the pathogenesis of PCOS. Furthermore, the use of ERS has been shown to improve various pathological manifestations of PCOS [3, 27]. We identified several target genes of PCOS, including USP25, YAP1, SIRT2, and SNRPN [18, 28–30]. These findings suggest a close association between abnormal alternative splicing events and the occurrence of PCOS.

## Consequence of isoform switch between Normal and PCOS granulosa cells

The rMATS method focuses on detecting systematic changes in alternative splicing under different conditions, yet struggles to elucidate the biological significance of these changes. To better understand the biological implications, we incorporated the isofomSwitchAnalyzeR package to predict alternative splicing events based on transcript usage. Upon comparing genes with differential alternative splicing changes identified by both methods, we found that only 260 genes (9.2%) were detected by both (Fig 3A). The FPKM value of the 260 genes can be found in S2 Table. The isoformSwitchAnalyzeR package suggests that isoform switching is driven by three biological mechanisms: alternative transcription start site (aTSS), alternative transcription termination site (aTTS), and alternative splicing (AS), with AS encompassing exon skipping (ES), intron retention (IR), alternative 5' donor site (A5), alternative 3' acceptor site (A3), multiple exon skipping (MES), and mutually exclusive exons (MEE). When comparing the isoform pairs that changed in the Normal and PCOS groups, we observed 1967 isoform switch pairs as AS events, with aTSS events showing the most pairs (Fig 3B). As shown in Fig 3C, we compared the enrichments of alternative splicing patterns between Normal and PCOS groups and found that all splicing patterns, except for IR and MEE, were statistically significant (FDR < 0.05, Fishers exact test). In the results obtained from isoformSwitchanalyzeR, both ES and MES refer to exon skipping events, with ES representing single-exon skips and MES representing multi-exon skips, which corresponds to SE (skipped exon) event type in rMATS. We observed that exon skipping events were the most prevalent in both software analyses. Furthermore, we identified that several isoform switch consequences were either enriched or depleted in the transition from Normal to PCOS samples (Fig 3D). For example, compared to the Normal group, NMD (nonsense-mediated mRNA decay) sensitivity was significantly more frequent PCOS samples (FDR < 0.05, proportion test).

## The YAP1 protein in PCOS has a shorter ORF and an additional functional domain

YAP1 is among the 260 genes with different splice variants identified using two methods simultaneously and has long been known as a susceptibility gene for PCOS [31–33]. In Fig 4A, four distinct transcripts of the YAP1 gene are displayed. The use of the ENST00000531439.5_1 transcript (YAP1_439) increases in PCOS patients, while the use of the ENST00000526343.5_1 transcript (YAP1_343) decreases. YAP1_439 contains an additional WW (X2) protein domain compared to YAP1_343. The WW domain is responsible for interacting with numerous proteins containing PPxY motifs in the Hippo pathway (where P is proline, x is any amino acid, and Y is tyrosine). The presence of a single WW or double WW domain may affect the affinity and specificity of YAP1 during interaction with these PPxY motifs [34, 35].

The gene expression data in Fig 4B showed no difference in the gene expression of YAP1 between normal and PCOS groups, consistent with the DEG results obtained from DESeq2

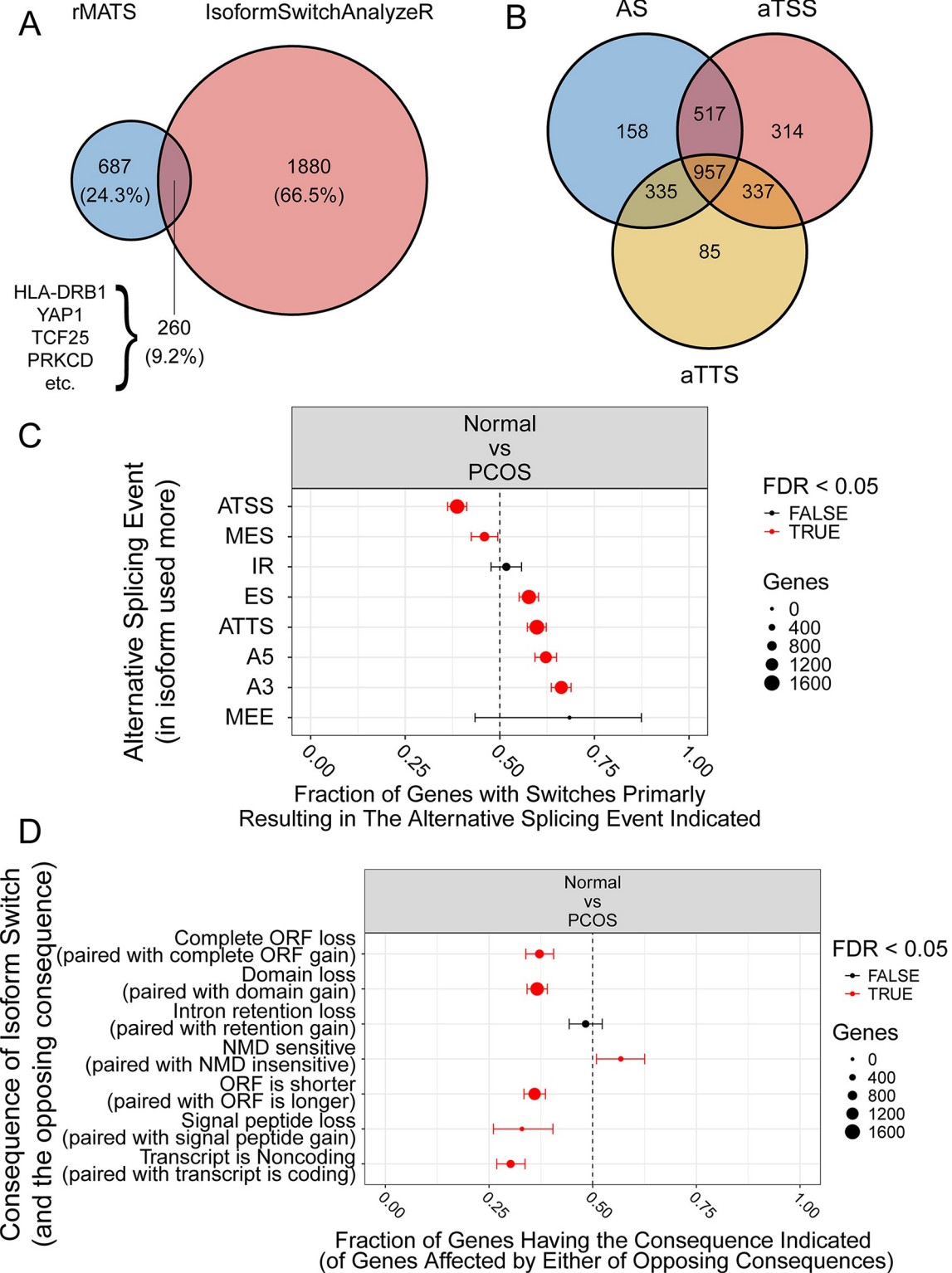

**Fig 3. Consequence of isoform switch between Normal and PCOS granulosa cells.** (A) Compare the genes that happened differential alternative splicing between rMATS and IsoformSwitchAnalyzeR. (B) The relationship among the isoform switch driven by three mechanisms, aTSS (alternative transcription start site), aTTS (alternative transcription termination site), and AS (alternative splicing). (C) Enrichment/depletion in alternative splicing between Normal and PCOS granulosa cells. The fraction (and 95% confidence interval) of isoform switches (x-axis) resulting in gain of a specific alternative splice event (indicated by y-axis) in the switch from

normal to PCOS. Dashed line indicates no enrichment/depletion. Splicing abbreviation is listed below: ATSS, alternative transcription start site; MES, multiple exon skipping; IR, intron retention; ES, exon skipping; ATTS, alternative transcription termination site; A5, alternative 5' donor site; A3, alternative 3' acceptor site; MEE, mutually exclusive exons. (D) Enrichment/depletion in isoform switch consequences between Normal and PCOS granulosa cells. From isoform switches resulting in either gain/loss of a consequence, the x-axis shows the fraction (with 95% confidence interval) resulting in the consequence indicated by y-axis, in the switches from Normal to PCOS. Color indicate if FDR < 0.05.

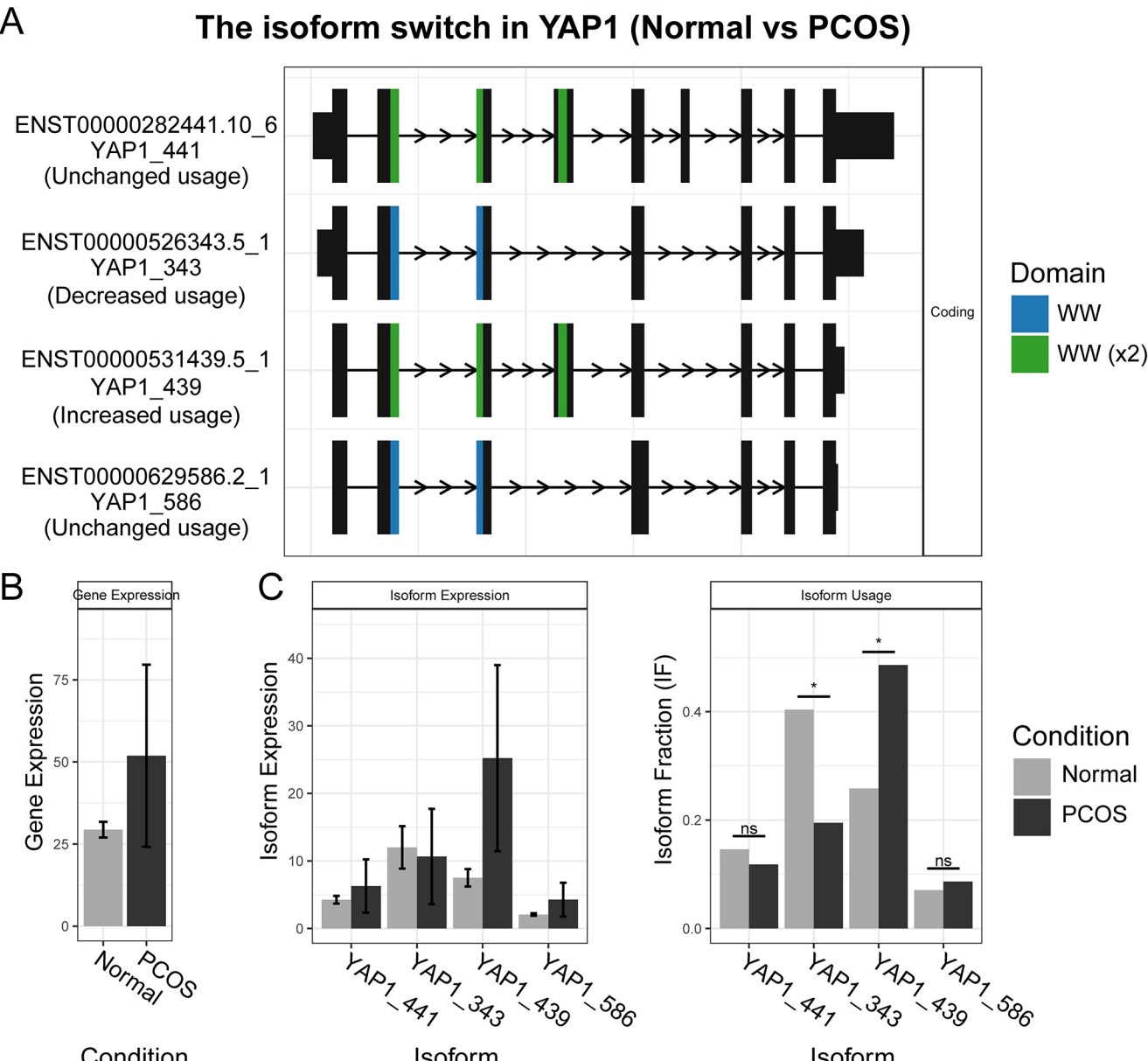

**Fig 4. The isoform switch in YAP1 between Normal and PCOS granulosa cells.** This plot illustrates the gene structure of different YAP1 transcripts and their expression and usage. (A) The gene structure and functional domain of each transcript. (B) The gene expression level of YAP1. (C) The isoform expression and usage of each transcript in YAP1.

(Fig 1A). However, a significant difference in the use of transcripts between the two groups was seen in Fig 4C. Taking into account, the alterations in transcript structure and usage, it can be inferred that YAP1_343 transcripts were predominantly utilized in the normal group, while in PCOS patients, the expression of YAP1_439 subtypes was substantially increased. This suggests the potential for YAP1 to interact with a new or different set of proteins in the PCOS environment.

## Discussion

Transcriptional profile is the mainstream analysis of RNA sequencing data. In this study, we first identified the differentially expressed genes (DEGs) which were found to be significantly enriched in immune related Gene Ontology (GO) terms, including those associated with inflammatory, interleukin and immune response. It is important to note that the number of DEGs identified in our analysis differs from that reported in the original study, due to differences in the filtering criteria applied. This is to some extent in accordance with the viewpoint that PCOS is not only a reproductive endocrinological disease but also a systematic disorder accompanying with long-term chronic inflammation [36], through which serious additional illness could be triggered such as cardiovascular issues [37]. Numerous studies showed potentially potent therapeutic efficacy with several anti-oxidants such as resveratrol [38], melatonin [4], ginger and cinnamon [39]. The association of inflammation and PCOS is well recognized but how they interact with each other or how the systemic effect of inflammation contributed to the process of this disease is still an open question. Our results hint at the involvement of ovarian granulosa cells in the low-grade inflammation response in PCOS patients.

We found genes with differential AS patterns are specifically associated with endoplasmic reticulum stress (ERS). In previous researches, ERS is triggered by abnormal situations such as hypoxia or imbalanced metabolic status, resulting in the unfolded protein response (URP) to recover the cellular homeostasis. Long-lasting ERS have been reported to associated with pathological conditions including PCOS [40]. ERS has been demonstrated to play important roles in the development and function of the ovary, and it is believed that proper level of ERS could be helpful for oocyte maturation [41]. The hyperandrogenism follicular microenvironment in PCOS activates the ERS in ovarian GCs through NK-κB, Notch2 [42] and inflammasome signaling, which could lead to pyroptosis [43] or apoptosis [41] in GCs and chronic inflammation in the ovary. Our results could be a potential bridge to connect PCOS, inflammation and ERS. It is of great importance to further investigate the detailed relationship and potential interaction of them to the development of novel pharmaceutical strategy aiming at ERS or inflammation related molecules.

YAP1 (yes-associated protein 1) has been reported as a PCOS risk gene by GWAS studies [44] and a putative target of pharmaceutical treatment [45]. As a downstream member of Hippo pathway, YAP1 is involved in masses of biological process such as cell proliferation, cell differentiation, DNA repair and hormone regulation [46]. A recent study demonstrated a detailed cell proliferation regulation of MIGA2-Hippo/YAP1 pathway in ovarian granulosa cells [47]. The ovarian change in PCOS is reported to be similar in YAP1 activation phenotype [46]. We observed higher expression of the YAP1 protein's long transcript (ENST00000531439.5_1) in PCOS patients compared to the short transcript (ENST00000526343.5_1). The long transcript possesses an additional double WW domain, influencing its binding affinity and specificity to PPxY motifs. Our finding of the isoform switch in YAP1 might provide a putative mechanism of PCOS occurrence and progress, which could also be related to the disease risk SNVs in YAP1 gene. Previous studies have

demonstrated the ability to directly activate cellular phenotypes associated with disease progression, offering further support for this potential mechanism [20].

## Materials and methods

### Quality control and quantification of transcriptome sequencing data

We obtained raw RNA-Seq data of granulosa cells from PCOS patients and healthy donors from previous research [24]. The data can be accessed using the following accession number: GSE138518. After removing adaptors and filtering out low-quality sequences from the raw reads using fastp (version 0.23.4) [48], we mapped the remaining clean reads from each sample to the annotated human genome (GRCh37.primary.assembly.genome.fa downloaded from the GENCODE database) using STAR (version 2.5.3a) software [49] in two-pass mode with default parameters. Subsequently, we quantified gene expression and transcript expression using featureCounts (version 2.0.1) [50] and RSEM (version 1.3.1) [51] software with the output files from STAR, respectively. The featureCounts (version 2.0.1) software used the 'Aligned.sortedByCoord.out.bam' file of each sample for gene expression quantification with the following settings: featureCounts -t gene -f -O -s 2 -T 6 -F GTF -a gencode.v40lift37.annotation.gtf -o output.txt *Aligned.sortedByCoord.out.bam. Meanwhile, RSEM (version 1.3.1) software quantified transcript expression using the 'Aligned.toTranscriptome.out.bam' file of each sample with the following settings: rsem-calculate-expression–paired-end–alignments -p 5 -no-bam-output -q ${sample}Aligned.toTranscriptome.out.bam reference_name sample_name.

### Differential expression analysis

The unnormalized counts from featureCounts were assembled into a count matrix with R (version 4.3.2) and served as an input for DESeq2 package (version 1.42.1) [52] to identify differentially expressed genes (DEGs). Only the genes with $|\log_2\text{FoldChange}| > 2$ and adjusted p-value $< 0.05$ were considered as DEGs. The resulting DEGs were visualized by volcano plot. The annotations of gene type were obtained through the getBM function in the biomaRt package (version 2.58.2) [53].

### Differential alternative splicing analysis

The classification of differentially alternatively spliced genes (DASGs) was analyzed using rMATS (replicate Multivariate Analysis of Transcript Splicing, version 4.1.2) [25] to compare PCOS and normal granulosa cells. rMATS is a computational tool that detects differential alternative splicing events (DASEs) from RNA-Seq data. It can automatically identify and analyze DASEs corresponding to all major types of alternative splicing patterns. rMATS can recognize five types of DASEs: skipped exon (SE), alternative 5' splice site (A5SS), alternative 3' splice site (A3SS), mutually exclusive exons (MXE) and retained intron (RI). rMATS employs the exon inclusion level to measure the extent of alternative splicing events occurring within a sample. The exon inclusion level indicates the proportion of isoforms that include a particular exon. A differential alternative splicing event (DASE) is recognized when the difference in exon inclusion levels between two groups surpasses 0.1 ($|\Delta\text{IncLevel}| > 0.1$) and the false discovery rate (FDR) is below 0.05.

### Differential transcript usage analysis

To facilitate the comparison of differences in the utilization and potential functional alterations of distinct transcripts from the same gene, we employed the isoformSwitchAnalyzeR

package (version 2.2.0) [54] to conduct a comprehensive analysis of differential transcript usage (DTU). The transcript annotation file (gencode.v40lift37.transcript.fa) was retrieved from the GENCODE database to import RSEM quantification of transcripts. Subsequently, the isoformSwitchTestDEXSeq() function was utilized for DTU analysis, with the analyzeAlternativeSplicing() function employed to scrutinize alternative splicing events. Notably, the variance between isoforms involved in an isoform switch can arise from alterations in three distinct biological mechanisms: alternative transcription start site (aTSS), alternative transcription termination site (aTTS), and alternative splicing (AS). These mechanisms encompass six types of AS events: alternative 5' donor site (A5), alternative 3' donor site (A3), exon skipping (ES), multiple exon skipping (MES), mutually exclusive exons (MEE), and intron retention (IR). Furthermore, to explore the influence of these DTUs on protein function, we integrated additional information including functional domains (from the PFAM database) [55], signal peptides (assessed via SignalP 6.0 fast) [56], open reading frames, and protein-coding prediction (analyzed using CPAT version 1.2.4) [57] under the specified guidelines.

### Gene function enrichment and network analysis

Gene Ontology (GO) categories were classified by the clusterProfiler R package (version 4.10.1) [58] for the annotation and enrichment of target genes. The Benjamini & Hochberg method was used to correct the p-value of GO terms enrichment, and we considered the p-value less than 0.05 and adjusted p-value less than 0.2 to be statistically significant. To reduce redundancy in the enrichment results, we employed the simplifyEnrichment R package (version 1.12.0) [59], which clusters GO terms based on their semantic similarity. Specifically, we applied the binary cut method from this package to cluster the similarity matrix of the enriched terms. For each cluster, we visualized a high-frequency word cloud.

## Supporting information

**S1 Fig. Quality control of RNA-Seq data.** (A) Unique mapped ratio of samples between Normal and PCOS. (B) Principal component analysis of samples between Normal and PCOS.
(TIF)

**S2 Fig. Similarity heatmap of BP terms.** The cluster of BP terms is enriched based on differentially expressed genes.
(TIF)

**S1 Table. Differential alternative splicing events between human Normal and PCOS granulosa cells.** The results are evaluated by rMATS and the criteras for filtering differential alternative splicing events is |ΔIncLevel| > 0.1 and FDR < 0.05. A3SS, alternative 3' splicing site; A5SS, alternative 5' splicing site; MXE, mutually exclusive exons; RI, retained intron; SE, skipped exons.
(XLSX)

**S2 Table. FPKM expression values of 260 differential spliced genes were detected by both rMATS and isoformSwitchAnalyzeR.**
(XLSX)

## Author Contributions

**Conceptualization:** Ruizhi Feng, Congxiu Miao.

**Data curation:** Linlin Yang, Hui Miao, Na Li.

**Formal analysis:** Linlin Yang, Jianhua Chen.

**Funding acquisition:** Ruizhi Feng, Congxiu Miao.

**Methodology:** Jianhua Chen.

**Visualization:** Huilin Bi.

**Writing – original draft:** Linlin Yang, Jianhua Chen.

**Writing – review & editing:** Ruizhi Feng, Congxiu Miao.

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
