## [Decision Letter · Decision Letter 0]

5 Nov 2024

PONE-D-24-44286The landscape of alternative splicing in granulosa cells and a potential novel role of YAP1 in PCOS

PLOS ONE

Dear Dr. Feng,

Thank you for submitting your manuscript to PLOS ONE. After careful consideration, we feel that it has merit but does not fully meet PLOS ONE’s publication criteria as it currently stands. Therefore, we invite you to submit a revised version of the manuscript that addresses the points raised during the review process.

During your revision, please address the issues raised by your reviewers, particularly the questions raised by Reviewer 1 and the issue of the statistical cut-off used to calculate the results of gene ontologies raised by Reviewer 2. In addition, please address the clarity of writing noted by both reviewers.

We look forward to receiving your revised manuscript.

Kind regards,

Eric A. Shelden, Ph.D.

Academic Editor

PLOS ONE

2. Thank you for stating the following financial disclosure: [This research was financially supported by grants from National Natural Science Foundation of China (81971451, 31900605), Innovative and Entrepreneurial Team of Jiangsu Province (JSSCTD202144), Innovative and Entrepreneurial Talent Program of Jiangsu Province to Ruizhi Feng, and Natural Science Foundation of Shanxi Province (NSF201901D111325) to Congxiu Miao.]. Please state what role the funders took in the study. If the funders had no role, please state: "The funders had no role in study design, data collection and analysis, decision to publish, or preparation of the manuscript." If this statement is not correct you must amend it as needed. Please include this amended Role of Funder statement in your cover letter; we will change the online submission form on your behalf.

3. Please note that your Data Availability Statement is currently missing [the repository name and/or the DOI/accession number of each dataset OR a direct link to access each database]. If your manuscript is accepted for publication, you will be asked to provide these details on a very short timeline. We therefore suggest that you provide this information now, though we will not hold up the peer review process if you are unable.

Additional Editor Comments (if provided):

Reviewers' comments:

Reviewer's Responses to Questions

**Comments to the Author**

1. Is the manuscript technically sound, and do the data support the conclusions?

Reviewer #1: Yes

Reviewer #2: Partly

2. Has the statistical analysis been performed appropriately and rigorously? 

Reviewer #1: Yes

Reviewer #2: No

3. Have the authors made all data underlying the findings in their manuscript fully available?

Reviewer #1: Yes

Reviewer #2: Yes

4. Is the manuscript presented in an intelligible fashion and written in standard English?

Reviewer #1: Yes

Reviewer #2: No

5. Review Comments to the Author

Reviewer #1: Review report for the manuscript entitled “The landscape of alternative splicing in granulosa cells and a potential novel role of YAP1 in PCOS” by Yang and colleagues (PONE-D-24-44286)

In this study, RNA-seq data from a previous study of ovarian granulosa cells from PCOS patients and normal human ovaries by another group were re-analysed to gain a deeper understanding of splicing isoforms specific to PCOS. The authors identified a number of differential splicing events in the patients that were associated with endoplasmic reticulum and post-translational modification processes. Among these, the YAP1 gene showed differential alternative splicing isoforms, producing long and short protein isoforms, suggesting a role in the HIPPO pathway in PCOS disease progression.

Overall, the previously published data set would be worthy of re-analysis with a focus on splicing isoforms, and the differential isoforms of YAP1 are interesting. However, I have some concerns with the manuscript as follows.

1. The abstract should explain the context of the study, which is a re-analysis of a previously published dataset.

2. In the Introduction: "A large number of cis-elements, such as relative RNA binding proteins, and trans-elements, such as enhancers, are involved in this complicated process" (lines 80-81). Usually, cis-elements are sequence elements in mRNA and genomic DNA, while trans-elements are their binding protein and enzymes. Therefore, the terms cis- and trans-elements should be exchanged.

3. In the Introduction section, “We analyzed the global expression of a previously public dataset with the RNA sequencing of granulosa cells in PCOS and control samples.” (line 91-93). It is surprising that the authors do not refer to the previous study that reported the original dataset in this sentence, as the original dataset is the main resource of this study.

4. In the Results section, “We obtained transcriptome sequencing data of granulosa cells from both PCOS patients and normal donors from previous studies.” (line 102-103). Again, the previous study, which is the main source of this study, should be cited here. In addition, the characteristics of the granulosa cells used in this study (e.g. age, method of isolation, follicular stage, validation of cell type) should be clarified here.

5. In the Results section, “we first analyzed differential alternative splicing events (DASEs, |ΔIncLevel|>0.1 and FDR < 0.05)” (line 128-129). Please provide the definition of "IncLevel" (inclusion level) in the Materials and Methods section and clarify what this parameter means to help readers assess its significance.

6. In the Results section, “Five types of alternative splicing events can be identified by rMTAS,”. (line 131). rMATS is the name of a program for analysing alternative splicing in RNA-seq data, and this should be clarified. Also, the abbreviation rMATS should be spelled out the first time it appears in the manuscript.

7. In Fig 2A, “Intersection size”. What is this parameter? The intersection between what and what? Is it the number of genes or the number of splicing events? Please clarify. Also, the identity of the horizontal axis should be given.

8. In Fig 2C, “clusters”. Is this the result of k-means clustering? If so, please clarify. Also, what if the number of k, which is the key parameter of k-means clustering and should be clarified.

9. In Fig 2A and 3B, the authors use different approaches to analyse the overlap of DASEs (combination of pie chart and upset plots versus Venn diagram), and these analyses were compared. Why are these different methods when comparing the results by alternative splicing analysis programs rMATS and isoformSwitchAnalyzeR? I think this comparison should be done using the same analysis method.

10. In the Results section, “Analysis of AS events between the two groups revealed no difference between IR and MEE, with ES being the most genetically involved in the remaining events (Fig 3C), consistent with rMATS results (Fig 2A).” (line 184-186). I could not understand this conclusion. In Fig. 3C, "MEE gain" shows a large value of Fraction of Genes ... with a large variety, while "IR gain" shows a small value, which would not lead to the conclusion "no difference between IR and MEE". Furthermore, I do not understand the phrase "with ES being the most genetically involved in the remaining events" and how this is supported by the data shown in Fig. 3C. Regarding the phrase "consistent with rMATS results (Fig 2A)", how were these results consistent? What was compared and how?

11. In Fig 3C ad 3D, “Fraction of Genes…” appears on the X-axis of the graphs. Please clarify the definitions and how these parameters were calculated.

12. In the Results section, “The use of the ENST00000531439.5_1 transcript (YAP1_439) increases in PCOS patients” (line 209). The protein identities (e.g. YAP1_439 and YAP1_343) should be provided here and in Fig 4A and 4C.

13. In Fig 4A, the predicted protein products should be shown in schematic diagrams.

14. Data accession. The processed data including the differential splicing variants in this study should be deposited in the GEO.

Reviewer #2: This is an overall interesting and potentially valuable reanalysis of previously published data. However, I have a major concern: the statistical cutoff used by the authors for gene ontology analysis is stated as an FDR < .2, but this is not conventionally considered statistically significant, and I think the results should be recalculated using an FDR <.05 or even FDR < .01. (see line 334).

I also urge the authors to make their R code and bash scripts available on a suitable repository, both to aid other investigators in reproducing this work and to validate it.

Finally, there are also a very large number of places where additional clarification and editing is needed, I have listed some of them below.

Lines 49 to 52 needs editing and citations.

Lines 64 and 65 “cell” should be “cells”.

67 “involving” should probably be “involvement”

68. Period after the citation

68 to 71 is not a sentence

80 to 82, I think cis and trans elements are mixed up.

89 “have”, should maybe be “generate” or similar.

92 “public” should be “published. A citation should be added.

92: “shed light upon the” should be “provide a”.

103: Citation?

106: What was PCA conducted on? How many genes and how were they selected?

109: The current authors found a difference number of DEGs than the original researchers that provided the data used in the previous study. These differences should be discussed.

112: the authors state that they found 2618 entries in the biological process category using a query of 892 expressed genes. What does this mean and how was this possible?

114: Higher than what? Should this be “high”? How is higher defined?

Figure 1: The use of binary-cut is not found in the methods.

151: “happened” should be something else, maybe “displayed” or “contained”?

158: I think the word results should be deleted.

158: I am not sure how the clustering was accomplished. The statement “based on their expression level” might imply that clustering was conducted on genes that were highly, lowly, moderately, etc, expressed – is that correct? I encourage the authors to make their scripts available.

174: It’s “isoformswitchAnalyzeR”.

177: see above.

177: switching.

193-194. This sentence needs to be rewritten.

239: “a mount” should be something else: “numerous” or “several”, or just delete replace “with a mount” with “of”?

242 should be “inflammation contributed to”

243: Our results hint at the involvement of..

271: Sentence needs to be rewritten

273: An odd choice to end the paper on this sentence. I think ending it after the sentence ending on line 271 should be considered.

287: What is [sample]?

6. PLOS authors have the option to publish the peer review history of their article (what does this mean?). If published, this will include your full peer review and any attached files.

Reviewer #1: No

Reviewer #2: No

---

## [Author Response · Author response to Decision Letter 0]

15 Nov 2024

Dear PLOS ONE Editorial Team and Reviewers:

We are deeply grateful for the guidance and assistance you have provided for our article (Manuscript Number: PONE-D-24-44286). Your expertise and experience have greatly improved the quality and readability of our work. We truly appreciate the effort and patience you have put into the review process. According to your nice suggestions, we have made extensive corrections to our previous manuscript. The reviewer comments are in italic font below, and specific concerns have been numbered. Point-by-point answers to the two kind reviewers are listed below this letter:

Response to Reviewer #1:

Thank you very much for your thorough and detailed review. Your valuable comments and suggestions have greatly enhanced the clarity, depth and readability of our manuscript. We have categorized your feedback and addressed each point individually. The changes made in the manuscript are highlighted in blue and can be found in the ‘Revised Manuscript with Track Changes.docx’ file. Below, you will find our specific responses to your comments.

Q1：The abstract should explain the context of the study, which is a re-analysis of a previously published dataset.

Q3: In the Introduction section, “We analyzed the global expression of a previously public dataset with the RNA sequencing of granulosa cells in PCOS and control samples.” (line 91-93). It is surprising that the authors do not refer to the previous study that reported the original dataset in this sentence, as the original dataset is the main resource of this study.

Q4: In the Results section, “We obtained transcriptome sequencing data of granulosa cells from both PCOS patients and normal donors from previous studies.” (line 102-103). Again, the previous study, which is the main source of this study, should be cited here. In addition, the characteristics of the granulosa cells used in this study (e.g. age, method of isolation, follicular stage, validation of cell type) should be clarified here.

Answer 1: Thank you very much for your thorough and thoughtful review. We sincerely apologize for not providing clearer instructions in the areas you pointed out. We have now included the necessary explanations regarding the characteristics of granulosa cells, along with appropriate citations, in the relevant sections of the manuscript. These changes are highlighted in blue for your convenience (Lines 29, 91-93, and 102-103). We greatly appreciate your valuable feedback and attention to the detail. 

Q2: In the Introduction: "A large number of cis-elements, such as relative RNA binding proteins, and trans-elements, such as enhancers, are involved in this complicated process" (lines 80-81). Usually, cis-elements are sequence elements in mRNA and genomic DNA, while trans-elements are their binding protein and enzymes. Therefore, the terms cis- and trans-elements should be exchanged.

Answer 2: Regarding your comments on the use of ‘cis-elements’ and ‘trans-elements’, we sincerely apologize for the oversight. After careful review, we have made the appropriate revisions to ensure accuracy (Lines 80-81). The terms have been exchanged to better align with their correct definitions. We truly appreciate your carefully review again.

Q5: In the Results section, “we first analyzed differential alternative splicing events (DASEs, |ΔIncLevel|>0.1 and FDR < 0.05)” (line 128-129). Please provide the definition of "IncLevel" (inclusion level) in the Materials and Methods section and clarify what this parameter means to help readers assess its significance.

Q6: In the Results section, “Five types of alternative splicing events can be identified by rMTAS,”. (line 131). rMATS is the name of a program for analysing alternative splicing in RNA-seq data, and this should be clarified. Also, the abbreviation rMATS should be spelled out the first time it appears in the manuscript.

Answer 3: rMATS (replicate Multivariate Analysis of Transcript Splicing) is a software tool used to define and analyze the expression of alternative splicing events by calculating exon inclusion levels (noted as IncLevel). The exon inclusion level represents the proportion of isoforms that contain a specific exon. A splicing event is regarded as a differential splice event if the absolute difference in exon inclusion levels (noted as |ΔIncLevel|) between two groups exceeds 0.1 and the false discovery rate (FDR) is less than 0.05. We have made clarifications in the relevant sections of the Materials and Methods and Results sections of the main text (Lines 139-140 and 339-344). Thank you again for your suggestions, which helps to clarify the information for readers.

Q7: In Fig 2A, “Intersection size”. What is this parameter? The intersection between what and what? Is it the number of genes or the number of splicing events? Please clarify. Also, the identity of the horizontal axis should be given.

Q11: In Fig 3C ad 3D, “Fraction of Genes…” appears on the X-axis of the graphs. Please clarify the definitions and how these parameters were calculated.

Answer 4: Apologies for the misunderstanding. In Fig. 2A, we present a combination of an upset plot and a pie chart. The upset plot illustrates the types of differential splicing events present in each gene. The height of the upper black bars in the upset plot corresponds to the number of gene sets with highlighted black dots in each column. The colored bars on the right indicate the number of genes associated with each differential splicing event type. The pie chart, on the other hand, shows the percentage of differential alternative splicing events. We have now added a more detailed explanation in the figure legend (Lines 157-161).

Fig. 3C displays the fraction (and 95% confidence interval) of isoform switches (on the x-axis) that lead to the gain of a specific alternative splice event (indicated on the y-axis) in the transition from normal to PCOS. This ratio is calculated by dividing the number of instances of a particular alternative splicing gain event by the total number of that event. Similarly, Fig. 3D shows the fraction (with 95% confidence interval) of isoform switches that result in either a gain or loss of a consequence. The x-axis shows the fraction of switches resulting in the consequence indicated on the y-axis, in the transition from normal to PCOS. Earlier, we simplified the annotation of the y-axis to save space in the layout. However, after your helpful suggestion, we realized that this simplification might cause confusion for readers. As a result, we have revised Figs. 3C and 3D accordingly and included the updated explanation in the figure legends (Lines 218-222, 226-230).

Once again, thank you for your insightful suggestion, which has greatly contributed to improving the clarity and readability of our manuscript.

Q8: In Fig 2C, “clusters”. Is this the result of k-means clustering? If so, please clarify. Also, what if the number of k, which is the key parameter of k-means clustering and should be clarified.

Answer 5: Thank you very much for your helpful reminder. In Fig 2C, the K-means algorithm was used for clustering. To ensure the reproducibility of the results, we set the random seed to 2024 and determined the optimal number of clusters to be 5 based on the elbow method (see in the below picture). As a result, the FPKM expression matrix of genes with differential splicing was subsequently clustered into 5 groups for further analysis. We have added the relevant explanations in both the figure legend (Lines 165-169) and the image itself to clarify this process.

Q9: In Fig 2A and 3B, the authors use different approaches to analyse the overlap of DASEs (combination of pie chart and upset plots versus Venn diagram), and these analyses were compared. Why are these different methods when comparing the results by alternative splicing analysis programs rMATS and isoformSwitchAnalyzeR? I think this comparison should be done using the same analysis method.

Answer 6: Thank you for your thoughtful comment. We understand your concern regarding the use of different methods for comparing the results obtained from the alternative splicing analysis programs rMATS and isoformSwitchAnalyzeR. The reason we employed both approaches is that each software has its own strengths and limitations, and we wanted to provide a comprehensive view of the results. By including both sets of analyses, we believe we can present a more robust and reliable interpretation of the differential alternative splicing events (DASEs).

In fact, when we compared the results from both tools, we found that only 9.2% of the genes were identified as having differential splicing events by both rMATS and isoformSwitchAnalyzeR (Fig 3A). This indicates that the overlap between the two software packages is relatively limited, highlighting the importance of using multiple methods to identify DASEs. Additionally, isoformSwitchAnalyzeR integrates information such as signal peptides, functional domains, and open reading frames, which makes it easier to interpret the biological significance of differential splicing events. For this reason, in our subsequent in-depth analysis, we have placed greater emphasis on the results obtained using isoformSwitchAnalyzeR. We have clarified this point in the manuscript to explain why both approaches were used (Lines 190-192).

We hope this addresses your concern, and we appreciate your valuable feedback to help improve the clarity of our analysis."

Q10: In the Results section, “Analysis of AS events between the two groups revealed no difference between IR and MEE, with ES being the most genetically involved in the remaining events (Fig 3C), consistent with rMATS results (Fig 2A).” (line 184-186). I could not understand this conclusion. In Fig. 3C, "MEE gain" shows a large value of Fraction of Genes ... with a large variety, while "IR gain" shows a small value, which would not lead to the conclusion "no difference between IR and MEE". Furthermore, I do not understand the phrase "with ES being the most genetically involved in the remaining events" and how this is supported by the data shown in Fig. 3C. Regarding the phrase "consistent with rMATS results (Fig 2A)", how were these results consistent? What was compared and how?

Answer 7: Thank you for your thoughtful feedback and for raising these important points. We apologize for the confusion in the interpretation of our results. To clarify, as shown in Fig. 3C, we compared the enrichment of alternative splicing patterns between the Normal and PCOS groups and found that all splicing patterns, except for IR and MEE, were statistically significant (FDR < 0.05, Fisher’s exact test). In the results obtained from isoformSwitchAnalyzeR, both ES (exon skipping) and MES (multiple exon skipping) refer to exon skipping events, with ES representing single-exon skips and MES representing multi-exon skips, which corresponds to the SE (skipped exon) event type in rMATS. We observed that exon skipping events were the most prevalent in both software analyses.

Furthermore, we identified that several isoform switch consequences were either enriched or depleted in the transition from Normal to PCOS samples (Fig. 3D). For example, compared to the Normal group, NMD (nonsense-mediated mRNA decay) sensitivity occurred significantly more frequently in PCOS samples (FDR < 0.05, proportion test).

To ensure clarity, we have rewritten this section in the manuscript to better explain our analysis and findings (Lines 202-212). We hope this revision addresses your concerns and enhances the clarity of our interpretation.

Q12: In the Results section, “The use of the ENST00000531439.5_1 transcript (YAP1_439) increases in PCOS patients” (line 209). The protein identities (e.g. YAP1_439 and YAP1_343) should be provided here and in Fig 4A and 4C.

Answer 8: Following your suggestion, we have updated Fig. 4A to include the corresponding protein identities beneath each transcript, and we have used these protein identities for labeling in Fig. 4C.

Q13: In Fig 4A, the predicted protein products should be shown in schematic diagrams.

Answer 9: We have listed several gene names in the Venn diagram of Fig. 3A and have uploaded the expression values of 260 differentially spliced genes, as detected by both methods, in Table S2 for your review.

Q14: Data accession. The processed data including the differential splicing variants in this study should be deposited in the GEO.

Answer 10: We greatly appreciate your valuable suggestion. In the interest of fostering better sharing and communication, we have uploaded the analyzed differential alternative splicing events as attachments. Additionally, the scripts and related resources used in this project are now available on our GitHub repository, which can be accessed at https://github.com/Jenny-chen98/ASinPCOS. We hope this will be helpful, and we sincerely welcome any further feedback or questions.

We are truly grateful for your thorough and thoughtful review of our manuscript. The careful attention you gave to every aspect of the manuscript has significantly contributed to its improvement, and we are genuinely thankful for the clarity and depth of your suggestions. 

Response to Reviewer #2:

Thank you for your constructive feedback on our manuscript. We sincerely appreciate your careful review of the statistical methodologies employed in our study. In addition, we acknowledge your suggestion regarding the need for further clarification and editing in several sections. We have addressed the issues you raised and made the necessary revisions to improve the clarity and precision of the manuscript. These changes are marked in blue text in the “Revised Manuscript with Track Changes.docx.”

We will now respond to your comments point by point, as listed below:

Q1: I have a major concern: the statistical cutoff used by the authors for gene ontology analysis is stated as an FDR < .2, but this is not conventionally considered statistically significant, and I think the results should be recalculated using an FDR <.05 or even FDR < .01. (see line 334).

Answer1: Thank you for your valuable feedback. The FDR threshold used in our gene ontology analysis was initially set to 0.2 to allow for a broader exploration of potential enriched terms and to observe a more comprehensive set of results. FDR is a statistical method that helps control for false positives when adjusting p-values. However, based on your suggestion, we recalculated the analysis using a more stringent q-value cutoff of 0.05 in the enrichGO function of the clusterProfiler package. Interestingly, the results were identical to those obtained with the FDR threshold of 0.2. This consistency further supports the robustness of our findings. 

Additionally, we have uploaded all the GO enrichment tables derived from the clustering in Figure 1C and Figure S2 to our GitHub repository, which can be accessed via the following link: https://github.com/Jenny-chen98/ASinPCOS/tree/main/results/01_DEG. As shown in these tables, all terms have a q-value of less than 0.05.

Q2: there are also a very large number of places where additional clarification and editing is needed, I have listed some of them below.

Lines 49 to 52 needs editing and citations.

Lines 64 and 65 “cell” should be “cells”.

67 “involving” should probably be “involvement”

68. Period after the citation

68 to 71 is not a sentence

80 to 82, I think cis and trans elements are mixed up.

89 “have”, should maybe be “generate” or similar.

92 “public” should be “published. A citation should be added.

92: “shed light upon the” should be “provide a”.

103: Citation?

106: What was PCA conducted on? How many genes and how were they selected?

109: The current authors found a difference number of DEGs than the original researchers that provided the data used in the previous study. These differences should be discussed.

112: the authors state that they found 2618 entries in the biological process category using a query of 892 expressed genes. What does this mean and how was this possible?

114: Higher than what? Should this be “high”? How is higher defined?

Figure 1: The use of binary-cut is not found in the methods.

151: “happened” should be something else, maybe “displaye

---

## [Decision Letter · Decision Letter 1]

1 Dec 2024

The landscape of alternative splicing in granulosa cells and a potential novel role of YAP1 in PCOS

PONE-D-24-44286R1

Dear Dr. Feng,

We’re pleased to inform you that your manuscript has been judged scientifically suitable for publication and will be formally accepted for publication once it meets all outstanding technical requirements.

Kind regards,

Eric A. Shelden, Ph.D.

Academic Editor

PLOS ONE

Additional Editor Comments (optional):

Reviewers' comments:

Reviewer's Responses to Questions

**Comments to the Author**

1. If the authors have adequately addressed your comments raised in a previous round of review and you feel that this manuscript is now acceptable for publication, you may indicate that here to bypass the “Comments to the Author” section, enter your conflict of interest statement in the “Confidential to Editor” section, and submit your "Accept" recommendation.

Reviewer #1: All comments have been addressed

2. Is the manuscript technically sound, and do the data support the conclusions?

Reviewer #1: Yes

3. Has the statistical analysis been performed appropriately and rigorously? 

Reviewer #1: Yes

4. Have the authors made all data underlying the findings in their manuscript fully available?

Reviewer #1: Yes

5. Is the manuscript presented in an intelligible fashion and written in standard English?

Reviewer #1: Yes

6. Review Comments to the Author

Reviewer #1: I have carefully reviewed the revised manuscript and found that the authors have addressed all my concerns. Therefore, I am pleased to recommend this paper for publication in PLOS ONE.

7. PLOS authors have the option to publish the peer review history of their article (what does this mean?). If published, this will include your full peer review and any attached files.

Reviewer #1: No

---

## [Editor Report · Acceptance letter]

3 Dec 2024

PONE-D-24-44286R1 

PLOS ONE

Dear Dr. Feng, 

I'm pleased to inform you that your manuscript has been deemed suitable for publication in PLOS ONE. Congratulations! Your manuscript is now being handed over to our production team.

Kind regards, 

on behalf of

Dr. Eric A. Shelden 

Academic Editor

PLOS ONE